# Mechanistic Interplay between HIV-1 Reverse Transcriptase Enzyme Kinetics and Host SAMHD1 Protein: Viral Myeloid-Cell Tropism and Genomic Mutagenesis

**DOI:** 10.3390/v14081622

**Published:** 2022-07-26

**Authors:** Nicole E. Bowen, Adrian Oo, Baek Kim

**Affiliations:** 1Department of Pediatrics, School of Medicine, Emory University, Atlanta, GA 30329, USA; nicole.eileen.bowen@emory.edu (N.E.B.); adrian.oo@emory.edu (A.O.); 2Center for Drug Discovery, Children’s Healthcare of Atlanta, Atlanta, GA 30329, USA

**Keywords:** retrovirus, human immunodeficiency virus type 1, reverse transcriptase, antiretroviral therapy, mutation, drug resistance, SAMHD1, cell tropism

## Abstract

Human immunodeficiency virus type 1 (HIV-1) reverse transcriptase (RT) has been the primary interest among studies on antiviral discovery, viral replication kinetics, drug resistance, and viral evolution. Following infection and entry into target cells, the HIV-1 core disassembles, and the viral RT concomitantly converts the viral RNA into double-stranded proviral DNA, which is integrated into the host genome. The successful completion of the viral life cycle highly depends on the enzymatic DNA polymerase activity of RT. Furthermore, HIV-1 RT has long been known as an error-prone DNA polymerase due to its lack of proofreading exonuclease properties. Indeed, the low fidelity of HIV-1 RT has been considered as one of the key factors in the uniquely high rate of mutagenesis of HIV-1, which leads to efficient viral escape from immune and therapeutic antiviral selective pressures. Interestingly, a series of studies on the replication kinetics of HIV-1 in non-dividing myeloid cells and myeloid specific host restriction factor, SAM domain, and HD domain-containing protein, SAMHD1, suggest that the myeloid cell tropism and high rate of mutagenesis of HIV-1 are mechanistically connected. Here, we review not only HIV-1 RT as a key antiviral target, but also potential evolutionary and mechanistic crosstalk among the unique enzymatic features of HIV-1 RT, the replication kinetics of HIV-1, cell tropism, viral genetic mutation, and host SAMHD1 protein.

## 1. Introduction

Retroviruses are a diverse family of viruses, which harbor two identical copies of a single-stranded RNA genome [1]. The hallmark replication strategy of retroviruses involves the reverse transcription of the viral RNA into double-stranded DNA prior to integration into the host cellular genome [1,2]. Among the different genera classified under the *Retroviridae* family, the *Lentivirus* genus, which includes human immunodeficiency virus (HIV-1 and HIV-2) and simian immunodeficiency virus (SIV), result in immunocompromised conditions, such as the HIV-associated Acquired Immunodeficiency Syndrome (AIDS), following an infection [1,3]. Activated CD4+ T cells and myeloid lineage cells, such as macrophages and microglia, generally serve as natural target cells of lentiviruses, in which viral reverse transcriptase (RT) catalyzes the chain of events leading to eventual proviral DNA integration [4,5]. The RT enzyme kinetics and cellular availability of dNTP substrates, which directly correspond to the virus-replication kinetics in myeloid cells, as well as the error-prone nature of lentiviral RT, have been an exciting topic of study in the past years. The lack of proofreading capabilities of lentiviral RTs is considered to contribute to the emergence of various mutations observed within the viral genome [6]. Furthermore, due to the primary role of RT in ensuring the successful replication of the invading virus particles, the enzyme has been established as a target for antiretroviral therapy (ART) regimens for HIV-1-infected patients. HIV-1 demonstrates productive infections in macrophages, albeit in the low deoxynucleoside triphosphate (dNTP) pool environment within the terminally-differentiated cells [5,7]. The limited availability of dNTP substrates for HIV-1 replication in macrophages is associated with the dNTP triphosphohydrolase (dNTPase) activity of a host factor known as SAM domain and HD domain-containing protein (SAMHD1) [8,9]. This dNTPase activity of SAMHD1, which is highly expressed in macrophages, contributes to its role as an HIV-1 restriction factor in myeloid cells. However, the lentiviruses that encode for the viral protein X (Vpx), such as HIV-2 and certain strains of SIV, do not succumb to the dNTPase activity of their host SAMHD1 proteins and replicate rapidly, even in macrophages [10,11]. This is due to the SAMHD1-counteracting effects of Vpx (and some Vpr), which targets the host protein for proteosomal degradation, resulting in the elevation of dNTP pools in the infecting macrophages [12]. Interestingly, recent studies have deciphered other biological significances of SAMHD1, apart from its negative regulation of HIV-1 replication. In this review, we aim to discuss the enzyme kinetics of HIV-1 RT and its mechanistic role in viral myeloid cell tropism and genetic mutation, in addition to development of ART inhibitors targeting the enzyme. We also elaborate on the regulation of SAMHD1 as a retroviral restriction factor and its evolution along with viral RT, as a result of the ever-going host–pathogen arms race.

## 2. Replication Kinetics of HIV-1 in Target Cells

Retroviruses must utilize intracellular dNTPs in order for viral RTs to generate proviral DNA from the RNA genome in infected cells. Lentiviruses are unique among the retrovirus family as they can infect both dividing (activated CD4+ T-cells) and non-dividing cell types (terminally differentiated macrophages). Dividing cells regularly synthesize new DNA during the mitotic S phase and therefore need more abundant dNTP concentrations. In fact, dNTP concentrations are ~200 times higher in activated CD4+ T cells (1–16 μM) than in macrophages (20–40 nM) [4,5]. The concentration of dNTPs in macrophages is below the *K_m_* of HIV-1 RT, resulting in slowed replication kinetics [13]. As a result, the DNA synthesis during infection of activated CD4+ T-cells occurs within 12–16 h, whereas it takes up to 36 h during macrophage infection [14]. This kinetic difference results in robust HIV-1 replication and the rapid cell death of infected activated CD4+ T cells, while infected myeloid cells are long-living and persistently produce low levels of the virus [15,16,17,18]. Accordingly, after CD4+ T-cell depletion in vivo, the infection is sustained by long-living infected macrophages, marking them as an important component in HIV-1 reservoirs [16,19]. Thus, the differences in viral replication kinetics established by varying intracellular dNTP pools have clear effects on the generation and maintenance of HIV-1 reservoirs. 

## 3. Structural and Enzymological Background of HIV-1 RT

Retroviral RT is the primary key factor in the successful replication of the virus following the infection of target cells, such as macrophages and CD4+ T cells. As retroviral replication involves the integration of proviral DNA into cellular genome, the viral single-stranded RNA must be converted into its complementary DNA following capsid disassembly in the host cytoplasm. Hence, the replication kinetics of pathogens from the *Retroviridae* family, including lentiviruses such as HIV-1, HIV-2 and SIV, heavily rely on their respective viral RTs. 

HIV-1 RT is made up of two structurally distinct 440-amino acid p51 and 560-amino acid p66 subunits, which are translated and assembled as part of the precursor Gag-Pol polyprotein (Pr160^Gag-Pol^) [20,21]. The final p51/p66 heterodimers are subsequently formed following the *Pol*-encoded protease-mediated proteolytic processing of Pr160^Gag-Pol^. The p66 subunit contributes to all three enzymatic properties of the viral RT, namely RNA-dependent DNA polymerase, DNA-dependent DNA polymerase, and RNase H activities. The DNA polymerase’s effects are found at the N-terminal domain of a hand-like structure consisting of the palm, thumb, and finger subdomains (Figure 1). Both DNA polymerase and RNase H domains are linked via a connection subdomain made up of residues 319–426 [22]. The p51 subunit, which was proposed to arise from the proteolytic cleavage of the immature p66 dimer, lacks RNase H activity as a result of the cleavage action at the p51-RNase H processing site [23,24]. Asp110, Asp185, and Asp186 are important in DNA polymerase activity [25,26,27], while Asp443, Glu478, Asp498, and Asp549 are residues important for RNase H activity [28,29]. On the other hand, the Y183 and M184 residues within the YMDD motif in the highly conserved polymerase-active palm subdomain of p66 are involved in regulation of dNTP binding within the RT-initiation complex (RTIC) [30]. 

RTIC is formed from the interaction of HIV-1 RT with viral genomic RNA (vRNA) and host-derived tRNA^Lys3^ primer, which binds to the primer-binding site of vRNA to initiate the RT-mediated synthesis of negative-strand DNA. A recent study reported a high-resolution RTIC, which was captured at 2.8 A [31]. In a similar manner to a previous study [32], the authors demonstrated a three-fold-higher nucleotide incorporation rate (*k_pol_*) when the p66 thumb domain of the HIV-1 RT was crosslinked with the primer–template complex prior to the polymerase reaction [31]. Although the resolved RTIC exhibited primer-extension activity, its cryo-EM structure produced contrasting findings, according to which the 3′ terminus of tRNA and the primer grip region of RT shifted away from the catalytic sites on the palm domain, indicating the inactive conformation of the RTIC. Various host and viral elements have been identified as cofactors required for the efficient processivity of the RTIC, and, therefore, for the initiation of viral genomic replication [33,34,35,36,37]. During HIV-1 replication, the positive-strand vRNA is non-specifically cleaved from the RNA–cDNA complex by the RNase H activity of viral RT, allowing the subsequent synthesis of second-strand DNA. The RNase H activity of HIV-1 RT also specifically removes the negative-strand tRNA^Lys3^ primer, as well as the viral polypurine tract (PPT) RNA primer, which is required to initiate the generation of the second-strand, positive-sense DNA.

## 4. HIV-1 RT Kinetics

The initiation phase of the DNA polymerase activity of HIV-1 RT was described as a ‘stop–start’ phenomenon, in which distinct pausing could be observed along the vRNA template prior to nucleotide elongation, which occurs at a more processive rate (about 50–500-fold higher than the initiation stage) [38,39]. The more sluggish performance of HIV-1 RT during initiation was due to the more rigid A-form wide structure of the vRNA, resulting in a hyperextended conformation of the RT within the RTIC [31]. The stem-loop structure of vRNA was also associated with RT pausing, which is characteristically observed during the early phase of initiation following the incorporation of the third nucleotide [40]. On the other hand, as reverse transcription progresses towards the elongation phase, the DNA–RNA hybrids and newly-synthesized dsDNAs develop into more flexible A- and B-form structures. 

Generally, HIV-1 RT exhibits more efficient polymerase activity (*K_m_*) under low-dNTP conditions when compared to other retroviruses, such as murine leukemia virus (MLV) and avian myeloblastosis virus (AMV) [5]. As relatively similar *k_cat_* and *k_pol_* values were determined between the HIV-1 and MLV RTs, the more promising *K_m_* observed with HIV-1 RT was associated with its higher binding affinity towards incoming nucleotides, *K_d_* (HIV-1 RT: 1.1 µM vs. MMLV RT: 40.2 µM) [5,41]. The ability of HIV-1 RT to function in the lower-dNTP environment allows the virus to replicate in terminally differentiated cells, such as macrophages. However, it is worth noting that HIV-1 RT kinetics vary depending on the templates’ nucleotide types. One of the earlier studies on this topic reported that the *K_d_* and *k_pol_* of HIV-1 RT against dATP were 4 µM and 33 s^−1^, and 14 µM and 74 s^−1^, when DNA and RNA templates were used, respectively [42]. Interestingly, structural and pre-steady-state kinetic analyses have shown that nucleotide incorporation during second-DNA-strand synthesis by the DNA polymerase domain occurs at a higher rate than the RNase H-mediated cleavage by viral RT (one cleavage activity for every seven nucleotides added to the new strand) [43,44]. This close coordination between both enzymatic activities of HIV-1 RT is especially crucial given the low nucleotide-incorporation efficiency during the initial stages of viral positive-strand DNA synthesis. 

The processivity of HIV-1 RT also highly depends on the concentrations of metal ions, such as Mg^2+^, during the reaction. HIV-1 RT fidelity has long been suggested to be underestimated due to the consistent use of higher-than-physiological Mg^2+^ in in vitro RT studies [45,46,47]. Nonetheless, the importance of the metal ions during a HIV-1 RT reaction was highlighted by the presence of two Mg^2+^ (3.6 A apart from each other) in the polymerase active site (D110, D185, and D186 residues), when the enzyme was in a complex with its substrates [30,48]. Low Mg^2+^ concentrations were attributed to increased polymerase activity, as well as decreased RNA degradation, by the RNase H of the enzyme, reducing the amount of fragmented DNA syntheses [49]. Specific interactions between Mg^2+^ and the 3′-OH moiety of incoming nucleotides were recently reported to contribute to the *k_pol_* of HIV-1 RT without having a significant influence on its nucleotide-binding affinity, *K_d_* [50]. When tested with Mg^2+^ concentrations close to the physiological level (0.5 mm, the reductions in the dependence of the reaction rate (*k_obs_*) were measured among dTTP analogs with 3′ modifications, in comparison to molecules harboring the 3′-OH. 

The polymerase kinetics of HIV-1 RT are also influenced by the presence of ATP, which dose-dependently decreases the initial velocity of dTTP incorporation into poly (rA)·p(dT)_12–18_ [51]. In addition to its suppressive effect on viral RT DNA synthesis efficacy, ATP also plays a role as a pyrophosphate donor that contributes to the excision of incorporated RT inhibitors, such as azidothymidine (AZT), enhancing the drug-resistance profiles of mutant RTs [51]. An increase in IC_50_ of more than 300-fold was computed when AZT-TP was exposed to a resistant mutant in the presence of ATP, whereas no difference in inhibitory potency was observed between wild-type and mutant RTs when ATP was absent in the reaction [51].

## 5. The Fidelity Profiles of HIV-1 RT

Approximately 59.7% of HIV-1 RNA, which primarily exhibits the G-to-A hypermutations (1 out of 127 reads), succumbs to RT-mediated nucleotide alterations in its genome [52,53]. The lack of 3′exonuclease proofreading properties in the viral RT [6] led to a progressive accumulation of mutations across different stages of the virus replication cycle (68% and 32% during the early and late stages, respectively) [54]. However, this remains a controversial claim, since the different steps of viral genomic replication require multiple proteins and enzymatic activities, including those involved in host and viral replication, such as host-DNA-dependent RNA polymerase II, host-DNA polymerase, and HIV-1 RT). There have been decades-long debates on whether the observed high-frequency variations in viral genomes sequenced from individual patients were actually due to the high-replication-turnover rate of the virus itself, or the lack of fidelity among the different enzymes involved (except the high-fidelity host-DNA polymerase). This remains an open discussion, as it is difficult to quantify the respective fractions of fidelity by host-RNA polymerase II or HIV-1 RT during viral replication, which contributes to the error rate of approximately 2 × 10^−5^ observed in the viral genome.

Nonetheless, kinetics and fidelity studies on HIV-1 RT have at least provided us with some informative numbers on the error rates of the viral enzyme itself. The base substitutions observed from nucleotide misincorporation by HIV-1 RT would result in the generation of a mismatch end of the primer-template during extension by the enzyme. Although HIV-1 RT exhibits relatively similar nucleotide-misinsertion fidelity to MuLV RT [41], the latter was proposed to possess a 15-fold-higher fidelity than HIV-1 [55,56,57]. This was explained by MuLV’s 3.8-fold reduction in binding affinity (*K_d_*) towards primer–template mismatch in comparison to that of HIV-1 RT [41]. As a result of the tendency of MuLV RT to detach from the primer–template mismatch, it harbors a much weaker extension efficiency (2.9 × 10^−4^ µM^−1^ s^−1^ vs. HIV-1 RT: 0.04 µM^−1^ s^−1^). Within the β3-β4 loop of the fingers subdomain of the HIV-1 RT p66 subunit, a few amino-acid residues have been associated with nucleotide misinsertion and mismatch primer-extension properties of the enzyme. Significant improvement in mismatch-extension fidelity was reported when the ddI-resistance-associated Leu74 (15–26-fold) and Val75 (3-fold) residues found within the dNTP binding pocket were mutated. Separately, lower levels of RNA product syntheses in the absence of either one of the dNTPs were detected when Lys65Arg mutation was introduced, and this was associated with the cumulative effects of changes in *k_pol_* and *K_d_* of the mutant RT, especially against misincorporated pyrimidines [58,59,60,61,62].

While, in general, a consensus has been reached regarding the highly mutagenic feature of HIV-1, varying mutation rates of the virus have been reported thus far, depending on the respective study designs. One of the earliest studies to demonstrate the variation observed in HIV-1 RT fidelity suggested that the mutation rates are highly dependent on the types of templates used [63]. The authors demonstrated that the enzyme was more error-prone (G-to-A substitutions) against the DNA template (30 × 10^−4^) than that of RNA (2.2 × 10^−4^). Interestingly, the error rate with the DNA template was about five-fold higher than in an earlier study, which reported an average of 5.9 × 10^−4^ forward mutant frequency using a recombinant HIV-1 RT bacteriophage, albeit both studies tested the enzyme’s fidelity using the *lacZ**α* gene in M13mp2 [6]. A separate study then compared the fidelity between the Klenow fragment of DNA Polymerase 1, MLV RT, and HIV-1 RT using DNA and RNA templates. Within the 80-base target, the HIV-1 RT generated mutation rates that were about five-fold higher in comparison to the other two enzymes, whereas relatively similar error rates were determined between both templates (1.7 × 10^−4^ for DNA vs. 1.5 × 10^−4^ for RNA) [64]. As the enzyme fidelity studies progressed further using HIV-1-specific genes, single-base substitutions were predominantly detected (76%) when a 153-base *env* V1 DNA sequence was subjected to 25 cycles of polymerase reactions using the viral RT [65]. Error rates ranging between 8.3 × 10^−6^ and 1.2 × 10^−4^ were calculated, depending on the types of substitutions; GC-to-AT transitions were the most commonly observed. 

Interestingly, the Hughes and Wu groups found that HIV-1 RT is only modestly more error-prone than RTs from retroviruses such as MLV, HTLV-1, BLV, SNV, and HIV-2 during single-cycle replication experiments [52,66]. However, these experiments utilized cancer cell lines, which harbor much higher dNTP levels than primary cells. As dNTP availability is a key factor in whether a polymerase extends after a misincorporation [67], thus generating a mutation, the high dNTP concentrations of these experiments likely account for why the RTs tested had similar error rates. We would expect single-cycle experiments conducted in primary macrophages where dNTPs are the limiting factor would result in a higher error rate for HIV-1 RT than for other RTs, as HIV-1 is more likely to extend a mismatched template/primer in these conditions [41].

HIV-1 RT fidelity also varies based on the specific viral genes copied, as well as the RNA structures of the coding regions. The viral DNA obtained from multiple infection cycles in HEK293T cells using a shuttle-vector-based assay revealed contrasting levels of accumulated point mutations between the *env* (3.6 × 10^−5^) and *int-vif-vpr* (7.5 × 10^−5^) coding regions [68]. A similar type of observation was also reported in the sequenced viral RNA extracted from EXP1293F cells transfected with vectors expressing the viral Gag, protease, p66 subunit of RT and codon-mutated RT [69]. Intriguingly, within the HIV-1 *env* coding sequence itself, the viral RT exhibited a 3.1-fold decrease in mutation rates at the V1–V5 regions of the outer apical domains compared to the remaining 60% parts of the gene [68]. The viral DNA isolated from the PBMC of severely infected patients also demonstrated similar mutational cold spots within the gp120 domains. 

## 6. Regulation of Cellular dNTP Pool as a Major Determinant of HIV-1 Replication Kinetics in Viral Target Cells

As described in the earlier sections of this review, the replication kinetics of HIV-1 or, more specifically, its viral RT, are greatly influenced by the dNTP concentration disparity between activated CD4+ T cells and macrophages. Cellular dNTP pools are established through a delicate balance of biosynthetic and degradation pathways. Two distinct pathways synthesize dNTPs, the de novo synthesis pathway and the salvage pathway. The de novo synthesis pathway generates dNTPs from NDPs, whereas the salvage pathway repurposes deoxyribonucleosides (dNs) from sources such as the DNA degradation of apoptotic cells into dNTPs [70,71]. A key enzyme in the de novo synthesis pathway is ribonucleotide reductase (RNR), which catalyzes the rate-limiting step in the reduction of the 2′ hydroxyl group on the sugar moiety of nucleoside diphosphate (NDP) substrates [70]. The activity of this enzyme is highly regulated by allosteric activation [72], cell-cycle-dependent expression [73,74,75], subcellular localization [76], and DNA-damage checkpoints [77]. This pathway is also dependent on the ability of nucleotide diphosphate kinase (NDPK) to phosphorylate the resulting dNDP to form the final dNTP product. The salvage pathway, however, is primarily composed of kinases, such as thymidine kinase 1 (TK1), which can mediate the transfer of phosphates to the dN substrate to generate dNTPs. Notably, several enzymes involved in dNTP biosynthesis are upregulated during the S phase to accommodate the dNTP needs of DNA synthesis [73,78].

Working in opposition to these dNTP-synthesis pathways is SAMHD1 [8]. SAMHD1 is a triphosphohydrolase that catalyzes the hydrolysis of cellular dNTPs into their dNs and triphosphate components [9]. The enzymatic activity of SAMHD1 establishes low-dNTP pools in cells such as macrophages, thus restricting HIV-1 replication kinetics at the reverse-transcription step [12,79,80]. SAMHD1 is made up of 626 amino-acid residues and it consists of three structural domains, an N-terminal nuclear-localization tag (^11^KRPR^14^) [4,81,82,83], followed by a SAM domain [84], an HD domain that houses the conserved histidine- aspartate residues that are critical for phosphohydrolase activity [5,84,85], and a C-terminal regulatory domain containing the T592 residue and two cyclin-binding motifs (^451^RXL^453^ and ^620^LF^621^), which are critical for regulation by the cell cycle [13,86,87,88]. SAMHD1 expression is consistent throughout the cell cycle, making post-translational modifications a critical regulator of the enzyme’s function [89]. 

## 7. Regulation of Human SAMHD1

Human SAMHD1 has several layers of post-translational modifications that regulate its dNTPase activity. First, it undergoes allosteric activation to form an enzymatically active tetramer [90,91] (Figure 2). Each monomer contains a catalytic site and two allosteric sites, A1 and A2 [92]. For activation, allosteric site 1 (A1) binds either dGTP or GTP to form the inactive dimer [93,94]. As GTP is more abundant in the low-dNTP-pool environment of macrophages, GTP serves as the primary activator in vivo [95]. The A2 site can accommodate any dNTP; however, the larger purine bases create more stable base–stacking interactions within the pocket and are therefore preferred in the order of dATP > dGTP > dTTP > dCTP [96]. The occupation of dNTPs in the A2 sites stabilizes the dimerization of dimers to form the tetramer structure [93]. In the active state, a Mg^2+^ bridges the phosphate groups of the dGTP/GTP of A1 and the dNTP of A2 [93,97]. Interestingly, it has been suggested that the stable tetramer can be maintained long after dNTP pools have been reduced to a level that would lead to inactivation (~10 µM) [98,99,100]. This property is likely to be essential for SAMHD1 activity in the low-dNTP-pool environments of non-dividing cells [5].

The catalytic site is able to accommodate any dNTP, as there is no specific interaction between the residues in the catalytic pocket and the base of the dNTP [92] (Figure 2). Instead, this interaction is mediated by a water network [92]. One implication of this promiscuity of the catalytic pocket is that SAMHD1 can hydrolyze dNTP analogues, such as ddNTPs, base-modified dNTPs, and cytarabine (ara-C) [95,101,102,103]. However, SAMHD1 cannot accommodate the 2′ hydroxyl group of rNTPs [95] or significantly hydrolyze NRTIs [104,105]. Recently, it was proposed that the catalytic site of SAMHD1 contains a bi-metallic Fe–Mg center [106], similar to the HD domains of other proteins [107,108,109]. In a structural analysis, it was found that these two metals are bridged by a water molecule, which dissociates to form a hydroxide ion and performs an in-line nucleophilic attack on the P^α^ [106]. 

The most extensively studied SAMHD1 post-translational modification is the CDK1/CDK2 CyclinA2-mediated phosphorylation of residue T592 [86,87,110,111] (Figure 2). This interaction is mediated by the two cyclin-binding domains of SAMHD1 [87,88] and corresponds with an increase in dNTPs before the S phase [88,112]. In non-dividing macrophages, this phosphorylation event is regulated by Raf/MEK/ERK-pathway activity [113]. The pSAMHD1 is dephosphorylated by the PP2A-B55^α^ holoenzyme during mitotic exit [89]. Molecular dynamics simulations indicate that the allosteric signal from the surface exposed T592 phosphorylation events are relayed to the core of the protein through the critical residues N452−K455 [114]. The results of this signal remain controversial. Some groups have found that pSAMHD1 destabilizes the tetramer, resulting in decreased triphosphohydrolase activity [97,100,115,116,117]. However, there are also conflicting reports that the dNTPase activity and tetramer stability of pSAMHD1 is comparable to SAMHD1 [110,118,119,120,121]. While phosphorylation may not regulate dNTPase activity, it appears to regulate HIV-1 restriction, as phosphomimic mutants lose their restriction capabilities [119,122].

SAMHD1 can be acetylated at residue K405 by arrest-defective protein 1 (ARD) [123] (Figure 2). This post-translational modification is seen at its most significant in the G1 phase and results in increased dNTPase activity [123]. Increased acetylation during the G1 phase may explain how SAMHD1 is able to lower dNTP pools to satisfy the G1 checkpoint and progress to the S phase [8,124]. Additionally, SAMHD1 undergoes SUMOylation at K595 residue [125] (Figure 2). This modification is dependent on both the SUMO-consensus motif (^592^TPQK^595^) at the site of SUMOylation and a proximal SUMO-interacting motif (SIM) (^488^LLDV^501^), which can contribute to the recruitment of SUMOylation machinery [125,126]. The inactivation of the SUMO consensus motif or the SIM suppresses HIV-1 restriction but does not impair dNTPase activity [125]. Interestingly, this was observed even when the T592 residue was dephosphorylated and SAMHD1 was expected to be antivirally active, which highlights the complexity of SAMHD1 antiviral regulation [125].

Finally, SAMHD1 can be regulated by the reversible oxidation of three surface-exposed cysteine residues, C341, C350, and C522 (Figure 2). Although C522 has been shown to have the highest redox activity, several X-ray crystallography studies of SAMHD1 observed the formation of C341–C350 disulfide bonds [9,90,96,127]. The Hollis group found that C522 acts as a primary sensor of redox signals and can form a disulfide bond with C341 or C350 that destabilizes tetramerization and abolishes dNTPase activity [128]. They also found that SAMHD1 is oxidized in cells in response to proliferation signals, which allow the accumulation of dNTPs [128]. C341A and C350A mutants showed decreased tetramerization and dNTPase activity, whereas C522A displayed phenotypes comparable to the wild type [128]. The Ivanov group obtained similar biochemical results with serine mutants, but observed wild-type dNTP depletion activity in vivo for all mutants [129]. Interestingly, the C522S and C341S mutations lose retroviral restriction, whereas the C350S is indistinguishable from the WT protein [129]. As with the T592 phosphorylation, this data shows a discrepancy between dNTPase activity, dNTP depletion, and HIV restriction. This story is further complicated by the molecular dynamic simulations performed by the Bhattacharya group, who indicated that C341S and C522S mutations cause drastic disruptions to the allosteric signaling network that extend to the catalytic site [130]. More recently, their in silico analyses suggested a role for these redox reactions in driving affinities for allosteric regulators, thereby fine-tuning the enzyme rather than turning it off or on [131]. The Hollis group recently proposed that phosphorylation of SAMHD1 causes a higher proportion of the enzyme to localize to the nucleus, thereby protecting it from oxidization [132]. Certainly, the regulation of SAMHD1 activity involves a tight interplay between nucleic acid binding, post-translational modifications, and redox action. 

## 8. HIV-1 Restriction by Human SAMHD1

SAMHD1′s dNTPase activity has the capacity to restrict HIV-1 at several steps in the viral life cycle. The low-dNTP-pool environment of macrophages established by SAMHD1 suppresses both the RNA- and the DNA-dependent proviral DNA synthesis of HIV-1 [5]. In these conditions, RT displays an increased strand-transfer frequency and is more reliant upon the central polypurine tract to complete proviral DNA synthesis [133,134,135]. Additionally, these low-dNTP conditions result in a 4–80-fold disparity between the rNTP/dNTP concentration ratio in macrophages compared to activated T cells [4]. This disparity drives an increase in ribonucleoside triphosphate (rNTP) and rNTP chain terminator incorporation during RT-mediated-DNA synthesis in macrophages [4,136]. The presence of an rNMP in the DNA template induces RT pausing, which can result in mutation synthesis [92]. While enzymes such as RNase H2 are usually able to repair rNMP misincorporations, RNase H2 activity is lower in macrophages than in dividing cells [136]. Furthermore, the RNase H2 repair process requires DNA-gap repair, which is dependent upon dNTPs and is therefore kinetically limited in non-dividing cells [137]. This also restricts HIV-1 at the integration step, as host-DNA polymerases need cellular dNTPs to repair the gap between single-stranded host DNA and partially integrated viral DNA [137,138]. Finally, SAMHD1-mediated low-dNTP pools of macrophages restrict endogenous reverse transcription (ERT), where partial reverse transcription occurs within cell-free viral particles utilizing co-packaged dNTPs [139]. HIV-1 virions produced by non-dividing cells harbor less dNTPs, consequently limiting ERT activity [139]. SAMHD1 dNTPase activity also restricts HIV-1 infection in other non-dividing cells, such as resting CD4 T cells [140,141] and dendritic cells [142,143]. Notably, SAMHD1 is able to suppress HIV-1 LTR-driven gene expression in dividing cells, and this feature is absent in both the phosphorylation (T592A) and catalytic site mutants [144]. However, this effect is more likely to be due to changes in nucleic-acid-binding ability than dNTPase activity, as dNTP levels are already high in this cell type [144]. Furthermore, the nucleic-acid-binding ability of SAMHD1 has recently been implicated in retroviral restriction [145]. 

## 9. Viral Protein X (Vpx) Counteracts Lentiviral Restriction by SAMHD1

HIV-2 and some SIVs can replicate rapidly, even in the macrophage environment, because they encode a viral accessory protein called viral protein X (Vpx), which is able to target SAMHD1 for proteasomal degradation [10,11,12,146]. Vpx accomplishes this by binding to the E3-ligase substrate adaptor, CUL4-associated factor 1 (DCAF 1), which creates a new surface that can recognize SAMHD1 [147]. This, in turn, complexes with DDB1, cullin 4, and ROC1/RBX1, followed by an E2 ligase in order to ubiquitinate SAMHD1 and target it for the proteasome [12,146,147,148,149,150,151]. This targeting of SAMHD1 by Vpx for proteasomal degradation occurs in the cell nucleus [81,82,83]. The transient degradation of SAMHD1 increases dNTP pools 10-fold in macrophages. As this is above the *K_m_* of RT, the reverse-transcription step occurs rapidly, and the frequency at which non-canonical rNTPs are incorporated during proviral DNA synthesis is lower [152]. This relieves Vpx-expressing viruses of the various restrictions SAMHD1 imposes, which allows SAMHD1-counteracting lentiviruses to replicate more efficiently in macrophages than SAMHD1-non-counteracting lentiviruses. Notably, Vpx originates from a gene supplication of viral protein R (Vpr) in an ancestral lentivirus [153,154,155] and, therefore, some Vpr proteins are able to target SAMHD1 for proteasomal degradation [148,156,157].

## 10. Host and Viral Proteins’ Evolution Due to the Host–Pathogen Arms Race

Restriction factors, such as SAMHD1, are engaged in an evolutionary arms race against viruses that succumb to their inhibitory properties [158,159]. The amino acids involved in the interaction between the restriction factor and its viral antagonist often display strong signatures of positive selection, and codons experience changes at a higher frequency than would be expected by neutral drift [160,161,162]. In fact, in old-world monkeys, SAMHD1 displays a positive selection signature in the N-terminal domain, whereas in new-world monkeys, SAMHD1 displays a positive selection in the C-terminal domain [156,163]. This discrepancy is dictated by the various requirements Vpx/Vpr orthologs have evolved for SAMHD1 recognition and degradation [164]. This evolutionary arms race between SAMHD1 and lentiviruses has also affected the evolution of viral RTs. Indeed, RTs from lentiviruses that cannot counteract SAMHD1 (such as HIV-1) polymerize DNA in macrophage conditions more efficiently than RTs from lentiviruses that do counteract SAMHD1 (such as HIV-2) [13,165]. The RTs from SAMHD1-non-counteracting viruses are more efficient because they have evolved to execute the conformational change step (*k_conf_)* more rapidly during the incorporation of dNTP molecules [166]. This increased efficiency allows the virus to bypass SAMHD1 restriction and replicate in the low-dNTP-pool conditions of macrophages [166]. Furthermore, RTs from SIVmac239 (Vpx-) infections acquire numerous amino-acid mutations that result in enhanced RT kinetics compared to RTs from SIVmac239 (Vpx+) infections [167]. This suggests that the loss of Vpx during an in vivo SIVmac239 infection can drive RT variations to counteract the limited availability of dNTPs in macrophages [167].

## 11. SAMHD1 and Host Innate Immunity

Mutations in the SAMHD1 gene were first associated with Aicardi–Goutières syndrome (AGS), a rare form of congenital neuropathy characterized by aberrant type 1 interferon (IFN) responses [168,169]. AGS patients develop hyperactivation of the innate immune response in the absence of infection, which interferes with brain development and causes death at early ages [168,170]. As proposed for other AGS-related proteins, such as TREX1 [171,172,173], RNase H2 [174], and ADAR [175], SAMHD1 mutations may interrupt cellular-nucleic-acid metabolism, which can allow native nucleic acids to accumulate and trigger the innate immune response through pattern-recognition receptors (PRRs) [176]. Another proposed mechanism for SAMHD1-mediated IFN downregulation suggests that the RNA- and DNA-binding properties of SAMHD1 may impede the RIG-I/MDA5 and cGAS/STING-mediated sensing of nucleic acids [145,177,178,179,180,181]. Additionally, SAMHD1 has been observed to directly downregulate the innate immune response by reducing the phosphorylation of the NF-κB inhibitory protein, IκBα, and reducing inhibitor-κB kinase ε (IKKε)-mediated IRF7 phosphorylation, inhibiting NF-κB and IRF7 activation [182]. In non-diving cells, this downregulation appears to be dependent on the dNTPase activity of SAMHD1; however, the effect is dNTPase-independent in dividing cells [182,183]. Further studies have demonstrated that SAMHD1-mediated NF-κB inhibition occurs through the TRAF6-TAK1 axis [184].

The ability of a virus to evade the innate immune response is critical to establishing a productive infection. A recent review highlights the many strategies HIV-1 employs to subvert the innate immune response, including counteracting restriction factors, disrupting signal-pathway transduction, and masking pattern-associated molecular patterns (PAMPS) [185]. Some studies have proposed that the restricted replication kinetics of HIV-1 in macrophages fall below the threshold that would trigger an interferon response [90,93]. This would suggest an evolutionarily favorable reason for HIV-1 to have lost the Vpx gene and the ability to counteract SAMHD1 [186]. Interestingly, as SAMHD1 suppresses innate immune responses [170,176,181], viruses such as SARS-Cov-2 are restricted in Vpx-treated cells or SAMHD1 knock-out cells due to the hyperactivated IFN environment [187]. The question of how Vpx-coding lentiviruses overcome this effect is under investigation. Some studies have observed that Vpx itself is able to suppress the innate immune antiviral response in monocyte-derived macrophages and monocytic cell lines [188,189]. One proposed mechanism is that Vpx is able to bind to the p65 subunit of NF-κB in order to suppress NF-κB activation [189]. This antagonism was conserved amongst Vpx proteins from distantly related lentiviruses and Vpr from SIV_mon_, which has SAMHD1-degradation activity [189]. This study is especially enticing, considering one suggested mechanism for SAMHD1-mediated immune suppression is at the NF-κB activation level [183,190]. Conversely, another study observed that Vpx elevated the innate immune response in macrophages independently of SAMHD1 [191]. The exploration of how SAMHD1 counteracting lentiviruses suppress or evade the hyperactive immune response triggered by SAMHD1 degradation, which likely involves a network of mechanisms, is an ongoing effort.

## 12. SAMHD1: Other Considerations

It is essential for cells to regulate dNTP pools in order for sufficient and balanced pools exist for DNA replication. Irregular or imbalanced dNTP pools result in lowered replication fidelity and an increase in mutation synthesis [192,193,194,195]. Aberrant dNTP pools can also sustain the uncontrolled/rapid cell division and higher cell population at the S phase of cancer cells [124]. In fact, elevated dNTP pools serve as a biochemical markers for cancer cells as they harbor 6–11-fold-higher dNTP levels than normal cells [196]. Interestingly, SAMHD1 mutations have been identified in a variety of cancers, including leukemias [197,198,199,200,201], lymphomas [202,203], lung cancer [204], and colon cancer [205,206,207]. Recently, a structural and functional investigation of two cancer-associated SAMHD1 mutants showed that the R366C/H mutants possessed protein-stability profiles similar to the wild type, and their only functional deficit was a dNTPase deficiency [208]. This suggests that SAMHD1 mutations contribute to the elevation of intracellular dNTP levels commonly observed in cancer cells [208] and mechanistically connects the respective roles of SAMHD1 in cancer and HIV-1 restriction [208]. Conversely, SAMHD1 expression can interfere with the anti-cancer efficacy of the triphosphorylated forms of some anti-cancer nucleoside analogues due to the enzyme’s ability to hydrolyze these compounds [101,104,209,210]. SAMHD1 expression has been shown to regulate the response of cancer cells to cytarabine (ara-C) [103,211], arabinosylguanine (AraG) [212], and 2′-C-cyano-2′-deoxy-1-β-D-arabino-pentofuranosyl-cytosine (CNDAC) [213]. Several suggestions have been made to target SAMHD1 in order to improve nucleoside-based anti-cancer therapies, including treatment with Vpx to degrade SAMHD1 [102] and treatment with RNR inhibitors to induce dNTP-pool imbalances and impede SAMHD1 allosteric activation [214].

SAMHD1 has other functions in cells, such as its involvement in DNA repair, where it localizes to sites of DNA damage [198,215]. SAMHD1 stimulates Mre11 endonuclease activity in order to promote the repair of double-strand DNA breaks and stalled replication forks [215,216]. Additionally, SAMHD1 dNTPase activity limits aberrant DNA resynthesis during non-homologous end-joining to repair double-strand DNA breaks [217]. This function has been shown to be important for DNA repair during antibody-class-switch recombination in B-cells [218]. Finally, SAMHD1 limits the accumulation of harmful DNA–RNA hybrids, called R loops, at sites of transcription-replication conflicts, where the DNA-replication machinery collides with the RNA polymerase, preventing replication fork stalling [219]. SAMHD1 is also involved in cell-cycle progression, as SAMHD1 knock-out cells accumulate in G1 phase of the cell cycle [124,220,221]. The additional features of these cells, such as proliferatory effects, depend on the cell type and were reviewed recently [222].

## 13. Perspectives

A significant barrier to the development of a cure for HIV is viral genetic diversity, both between patients and within each individual [223,224]. This high rate of genetic mutation allows HIV-1 to effectively escape from host immune responses and develop resistance to current antiretroviral treatments (ART). The reverse-transcription step as a result of the low replicative fidelity of viral RT is a key source of HIV-1 genomic mutagenesis [55,225], while other potential sources, such as deaminase-mediated mutagenesis [226,227] and the incorporation of non-canonical nucleotides [136,152], were also investigated.

In this review, we explained that the low-dNTP-pool environments of non-dividing target cells, such as macrophages, are established by the host lentivirus restriction factor, SAMHD1 [9,12,79,80]. This poor dNTP substrate availability can generate selective pressure to enable efficient DNA syntheses by HIV-1 RT, even at the low dNTP concentrations found in macrophages, via its uniquely low steady-state *K_m_* and pre-steady-state *K_d_* values (high dNTP binding affinity) compared to the RTs of retroviruses that infect only dividing cells harboring abundant dNTPs [13]. The unique enzymatic DNA-synthesis proficiency of HIV-1 RT in low-dNTP-level environments may contribute to the ability of the virus to infect myeloid cells with SAMHD1-mediated dNTP depletion. This possibility is further supported by the observation that HIV-2 and some SIVs, which counteract SAMHD1 via Vpx-protein-elevating dNTP pools [10,11] and rapidly replicate, even in myeloid cells, possess RTs with higher *K_m_* values than the RTs from SAMHD1 non-counteracting viruses, compared to HIV-1 RT [166]. Moreover, the RTs from SIVmac239 (Vpx-) infections acquire RT variants with enhanced polymerase activity compared to the RTs from wild-type SIVmac239 (Vpx+) infection [167]. Together, this suggests that the low-dNTP-pool environment established by host-SAMHD1 serves as a selective pressure for RT evolution. Specifically, SAMHD1-non-counteracting lentiviruses, such as HIV-1, may have evolved more efficient RTs in order to maintain macrophage tropism.

More broadly, we modeled how the error-prone nature of HIV-1 RT stems from the same selective pressure that drives higher polymerase efficiency (Figure 3). In low-dNTP-pool environments, polymerases are more likely to pause, which can generate mutation hot spots [195]. In order to survive and generate progeny in macrophages in which SAMHD1 is present, the virus must complete reverse transcription regardless of the likely increase in mutation synthesis due to the limited availability of dNTPs. In addition, HIV-1 RT exhibits higher primer-mismatch extension capability after misinsertion, compared to RTs of retroviruses that infect only dividing cells [41]. Possibly, the highly unique dNTP-binding affinity and efficient DNA-synthesis capability of HIV-1 RT, which enables the virus to infect macrophages, can also enhance its ability to extend mismatched primers after nucleotide misincorporation. We propose that the unique, high-dNTP incorporation capability of HIV-1 RT and its efficient extension from mismatched primers are mechanistically connected with viral myeloid-cell tropism and SAMHD1. This subsequently contributes to the error-prone nature of HIV-1 RT, and mechanistically supports the development of the high mutation rate of HIV-1.

Furthermore, as a result of its efficient mismatch-extension capability, HIV-1 RT may be able to complete reverse transcription even after a mutation that would normally terminate or stall DNA synthesis, leading to the production of live, mutant viruses. Conversely, in the case of retroviruses containing RTs incapable of extending post-misinsertion, mismatch primers are likely to fail to produce new mutant virions, as the reverse-transcription step of these viruses would be halted upon mismisertion. Overall, it is highly plausible that the efficient dNTP-incorporation efficiency of HIV-1 RT, which was likely gained in order to escape from SAMHD1 restriction in myeloid cells, may simultaneously contribute to the high mutation rate of HIV-1.

Overall, in this review, we discussed the potential mechanistic and evolutionary connections among the unique enzymatic features of HIV-1 RT, HIV-1 myeloid-cell tropism, HIV-1 mutagenesis, and host SAMHD1 (Figure 3).

## Figures and Tables

**Figure 1 viruses-14-01622-f001:**
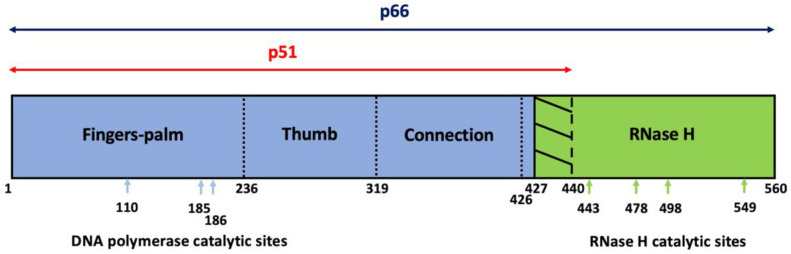
Schematic representation of HIV-1 RT structure. HIV-1 RT consists of two subunits, p66 and p51. The p66 subunit harbors both DNA polymerase (blue segment) and RNase H (green segment) domains, whereas p51, which is a proteolytic cleavage product of p66 subunit, does not have an active RNase H domain (residue 427–440). HIV-1 RT DNA polymerase domain, which captures DNA primer annealed to RNA or DNA template, in the form of a clasping right hand, can be further classified into the finger, palm, and thumb subdomains, whereas the connection subdomain links polymerase and RNase H domains. Amino-acid residues crucial for the enzymatic catalytic sites of DNA polymerase and RNase H activities are marked with respective colors.

**Figure 2 viruses-14-01622-f002:**
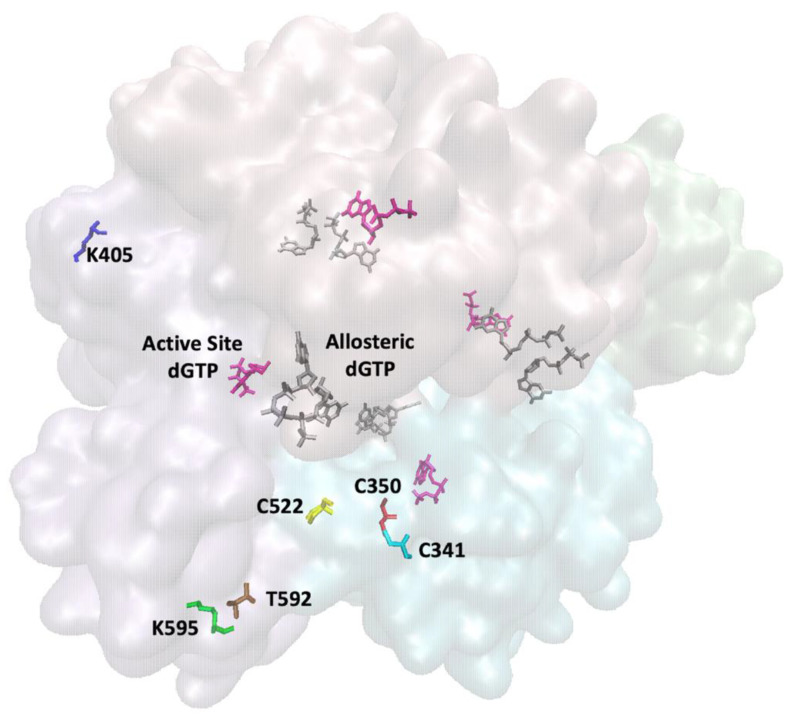
Structural representation of SAMHD1 regulation. The active SAMHD1 tetramer forms when each allosteric site is filled with nucleotides, shown above as dGTP (gray). This allows for the active site to bind deoxynucleotides for hydrolysis, shown above as dGTP (pink). SAMHD1 hydrolysis activity is regulated by several post-translational modifications: phosphorylation at T592 (brown), acetylation at K405 (blue), SUMOylation at K595 (green), and oxidation of C341 (teal), C350 (red), and C522 (yellow). Regulatory residues are shown for only one chain for model simplification. PBD structure 4BZB [93] and VMD software were used for modeling.

**Figure 3 viruses-14-01622-f003:**
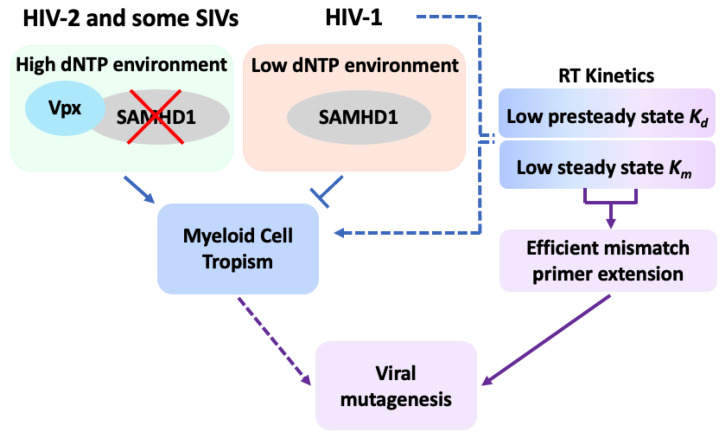
Model of mechanistic interplay among cellular dNTP levels, SAMHD1, RT enzyme kinetics, cell tropism, and genomic mutagenesis of HIV-1. In non-dividing cells, such as macrophages, dNTP pool is typically low as a result of the dNTPase activity of SAMHD1. Lentiviruses harboring Vpx, such as HIV-2, and several SIV strains counteract this antiviral restriction effect of SAMHD1 by proteasomally degrading SAMHD1 and elevating cellular dNTP levels, which promotes viral reverse-transcription kinetics in macrophages. By contrast, HIV-1 bypasses the low-dNTP pool of macrophages (dashed blue lines) via the unique kinetic properties of RT (low *K_d_* and *K_m_* values), which facilitate viral reverse transcription within this limited-dNTP cellular environment. This mechanistically contributes towards the myeloid-cell tropism of HIV-1. Concomitantly, this uniquely efficient dNTP incorporation efficiency of HIV-1 RT can also promote the high efficiency of HIV-1 RT in mismatched-primer extension, which can increase viral mutagenesis, suggesting that myeloid cell tropism and high genomic variability of HIV-1 are mechanistically connected (dashed purple lines).

## Data Availability

Not applicable.

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
