# Peer review of "Mechanistic Interplay between HIV-1 Reverse Transcriptase Enzyme Kinetics and Host SAMHD1 Protein: Viral Myeloid-Cell Tropism and Genomic Mutagenesis"

_viruses, 2022, doi:10.3390/v14081622_

Round 1

Reviewer 1 Report

In this manuscript, Oo, Bowen, and Kim attempt to argue for the existence of a mechanistic and evolutionary connection between the ability of HIV-1 to infect myeloid cells and genetic variation in HIV-1.  Specifically, the authors posit that the absence of the Vpx protein in HIV-1 (and hence, the inability of HIV-1 to counter SAMHD1-mediated restriction) resulted in a selective pressure that required HIV-1 to evolve a more efficient RT which could complete DNA synthesis under the low intracellular dNTP levels found in macrophages.  This adaptation, in turn, resulted in a more error-prone RT (relative to other retroviruses) that is responsible, at least in part, for the high genetic variability of HIV-1.  There are several problems with this idea, as discussed below.

The review also covers a broad swath of other concepts including the regulation of SAMHD1 enzymatic activity, resistance to RT inhibitors used in antiretroviral therapy, the role of SAMHD1 in innate immunity, and the involvement of SAMHD1 in DNA damage, cell cycle progression, and cancer.  As such, the review lacks focus, and many of the data and ideas described in these sections were discussed at length in a review by the same group that was published in Viruses in March of 2020 (vol. 12(4), 382; PMID: 32244340).

Major Comments:

1) The terms “hyper-mutagenesis” and genetic “hypermutability” are misleading, at best.  Hypermutations commonly refer to a specific class of clustered nucleotide changes such as those produced by APOBEC proteins, and mutability generally refers to the robustness of an organism (i.e., its ability to retain fitness in the face of mutations in a particular gene, or in the genome as a whole).  Instead, the authors seem to be pointing to the idea that mutations in HIV-1 occur at a higher rate relative to other retroviruses.  As explained in the next comment, there is nothing unique about mutation rate of HIV-1.

2) No mention is made of the numerous studies showing that the mutation rate of HIV-1, as measured in a single round of viral replication in culture, is on par with the mutation rates seen for other retroviruses including MLV, HTLV-1, BLV, and SNV.  In addition, a study of a lacZ-expressing HIV-1 vector by Abram et al. (J. Virol 2010; PMID: 20660205) concluded that HIV-1 replication is not especially error-prone relative to that of other retroviruses.  This finding, and the bulk of the data from single-round mutation rate experiments, suggests that in the context of the replicating virus, HIV-1 RT is not significantly less faithful than the RTs of other retroviruses.  More recently, Rawson et al. showed that, relative to HIV-1, HIV-2 (which expresses Vpx) shows only a slight reduction (<2-fold) in the overall frequency of transition mutations (transversions and insertions/deletions could not be measured due to technical limitations; J. Mol. Biol. 2017, PMID: 28502791).  Again, this points to the conclusion that HIV-1 RT is not particularly error-prone.

3) Even if we assume that HIV-1 RT has a “uniquely” low fidelity, several large leaps of logic are needed to directly tie macrophage infection to error-prone DNA synthesis.  First, one would have to assume that the selection pressure of HIV-1 replication in macrophages is sufficient to drive HIV-1 adaptation– which is a reach, since the primary target of HIV-1 in vivo is CD4-positive T cells.  Second, one would have to rule out the possibility that other selective forces, including CTL selection, are involved in shaping the biochemical properties of RT.  Third, one would have to make the argument (as the authors attempt to do) that adaptation towards a more efficient polymerase would result in poorer RT fidelity.  Much more work needs to be done with a broad array of retroviral RTs to rigorously test this idea.  Lastly, we are asked to assume that RT is not just a contributor, but the principal source of retroviral mutations.  This matter is still one of open debate.  Overall, the data that are presented by the authors as support for their hypothesis appear to be a series of loosely-related properties– they all center around RT, but there is insufficient evidence for a mechanistic or evolutionary link between them. 

4) Section 8.1 (lines 369-499) pertaining to RT inhibitors and drug resistance is tangential to the central theme, and several recent reviews, including one published in Viruses earlier this year, have covered this topic in detail.  See Singh and Das, Viruses 2022 (PMID: 35632767), Ruiz et al. Drug Discovery Today 2022 (35218925), and Cilento, Kirby, and Sarafianos Chem. Rev. 2021 (33507067).

5) Sections 8.2, 9, and 10 (lines 501-597).  Large portions of these three sections are simply a restatement of ideas expressed in the aforementioned 2020 review by the same research group (Coggins et al., Viruses 12(4), 382; PMID: 32244340).  The few pieces of relevant data that were published after the Coggins paper (i.e., lines 521-526 and 580-584) do not justify a new review article.

6) Section 12 (lines 629-684).  This section discusses the role of SAMHD1 in cancer, DNA damage, and cell cycle progression.  The information presented here does not fit the theme of HIV-1 macrophage infection and reverse transcriptase.  In addition, the concepts discussed in lines 656-673 were already covered in the review article mentioned in my previous comment.

Author Response

Reviewer 1:

1) The terms “hyper-mutagenesis” and genetic “hypermutability” are misleading, at best.  Hypermutations commonly refer to a specific class of clustered nucleotide changes such as those produced by APOBEC proteins, and mutability generally refers to the robustness of an organism (i.e., its ability to retain fitness in the face of mutations in a particular gene, or in the genome as a whole).  Instead, the authors seem to be pointing to the idea that mutations in HIV-1 occur at a higher rate relative to other retroviruses.  As explained in the next comment, there is nothing unique about mutation rate of HIV-1.

  • We thank the reviewer for pointing out this need for clarification. In our revised manuscript we have removed the phrase “hypermutability” and “mutability”. In their place we have clarified that we are discussing the high mutation rate of HIV-1.

2) No mention is made of the numerous studies showing that the mutation rate of HIV-1, as measured in a single round of viral replication in culture, is on par with the mutation rates seen for other retroviruses including MLV, HTLV-1, BLV, and SNV.  In addition, a study of a lacZ-expressing HIV-1 vector by Abram et al. (J. Virol 2010; PMID: 20660205) concluded that HIV-1 replication is not especially error-prone relative to that of other retroviruses.  This finding, and the bulk of the data from single-round mutation rate experiments, suggests that in the context of the replicating virus, HIV-1 RT is not significantly less faithful than the RTs of other retroviruses.  More recently, Rawson et al. showed that, relative to HIV-1, HIV-2 (which expresses Vpx) shows only a slight reduction (<2-fold) in the overall frequency of transition mutations (transversions and insertions/deletions could not be measured due to technical limitations; J. Mol. Biol. 2017, PMID: 28502791).  Again, this points to the conclusion that HIV-1 RT is not particularly error-prone.

  • We thank the reviewer for their feedback. We regret not discussing these papers and the implications of these data for our premise in our original manuscript. Both the Abram paper and the Rawson paper utilized cancer cell lines, HOS cells and U373-MAGI cells, respectively, for their experiments. Cancer cell lines contain significantly higher dNTP levels than cycling primary cells such as T-cells. This difference is even more drastic when considering macrophage dNTP levels. dNTP availability is a key factor in extending a mismatched primer/template and, as demonstrated in these publications, in high dNTP concentrations all of the RTs tested will extend the mismatch and generate a mutation equally. However, at the lower dNTP concentrations in macrophages, HIV-1 RT is uniquely able to extend after misincorporation to generate a mutation, whereas other RTs are unable to perform this extension to generate the mutation (Skasko et al. J. Biol Chem 2006; PMID: 15644314 ). Therefore, while useful, these infectious experiments would have to be done in the physiologically relevant cell types in order to observe the differences we see in biochemical experiments. We are grateful for the reviewer for pointing out this need for clarification and have added language to section 5 to address this matter.

3) Even if we assume that HIV-1 RT has a “uniquely” low fidelity, several large leaps of logic are needed to directly tie macrophage infection to error-prone DNA synthesis.  First, one would have to assume that the selection pressure of HIV-1 replication in macrophages is sufficient to drive HIV-1 adaptation– which is a reach, since the primary target of HIV-1 in vivo is CD4-positive T cells.  Second, one would have to rule out the possibility that other selective forces, including CTL selection, are involved in shaping the biochemical properties of RT.  Third, one would have to make the argument (as the authors attempt to do) that adaptation towards a more efficient polymerase would result in poorer RT fidelity.  Much more work needs to be done with a broad array of retroviral RTs to rigorously test this idea.  Lastly, we are asked to assume that RT is not just a contributor, but the principal source of retroviral mutations.  This matter is still one of open debate.  Overall, the data that are presented by the authors as support for their hypothesis appear to be a series of loosely-related properties– they all center around RT, but there is insufficient evidence for a mechanistic or evolutionary link between them. 

  • We thank the reviewer for their comments. Non-primate lentiviruses are considered the ancestral origin of primate lentiviruses. Many of these viruses, including equine infectious anemia virus (Sellon 1992 J. Virol PMID: 1382143), ovine lentivirus (Brodie 1995 Am J Pathol PMID: 7531949), visna-maedi virus (Genelman 1985 Proc Natl Acad USA Sci PMID: 2996004) and caprine arthritis-encephalitis virus (Zink 1990 Am J Pathol. PMID: 2327471) are mainly or exclusively macrophage trophic. Macrophage tropism is a more “intrinsic” property of the lentivirus and T-cell tropism was obtained later in evolutionary history. Therefore, it is likely that macrophage tropism posed an important selective pressure on the lentivirus RT. Additionally, our manuscript acknowledged the role of other sources of selection and mutation synthesis such as APOBEC, in the discussion section. We agree, of course, that much more work on retroviral RT must be done to rigorously test our perspective and come to a conclusion. However, we feel there is sufficient biochemical and cellular sufficient data to suggest a model for the field (Figure 3). In response to this feedback, we have added more language to the discussion section (now “Perspective”) to highlight the model is only an opinion based on current evidence.

4) Section 8.1 (lines 369-499) pertaining to RT inhibitors and drug resistance is tangential to the central theme, and several recent reviews, including one published in Viruses earlier this year, have covered this topic in detail.  See Singh and Das, Viruses 2022 (PMID: 35632767), Ruiz et al. Drug Discovery Today 2022 (35218925), and Cilento, Kirby, and Sarafianos Chem. Rev. 2021 (33507067).

  • We thank the reviewer for their comments. We have removed this tangential section on RT inhibitors and drug resistance in order to maintain the focus of the article.

5) Sections 8.2, 9, and 10 (lines 501-597).  Large portions of these three sections are simply a restatement of ideas expressed in the aforementioned 2020 review by the same research group (Coggins et al., Viruses 12(4), 382; PMID: 32244340).  The few pieces of relevant data that were published after the Coggins paper (i.e., lines 521-526 and 580-584) do not justify a new review article.

  • We thank the reviewer for their feedback. While the topics in sections 8.2 (now section 8), 9, and 10 have had only modest updates since the Coggins paper, we believe these sections are an essential part of our story-building for the premise of this review. Explaining SAMHD1 restriction, how Vpx overcomes this restriction, and the known effect of this host-virus interaction on the evolution of both the host and the virus establishes, in part, the feasibility of RT kinetics, mutagenesis, SAMHD1, and myeloid cell tropism being mechanistically connected. Furthermore, the Coggins paper is a comprehensive review of SAMHD1 and the disease impact of the enzyme. While our review shares some topics, our perspective is much different. However, we have made cuts to these sections and referred readers to the Coggins paper when appropriate in response to this feedback.

6) Section 12 (lines 629-684).  This section discusses the role of SAMHD1 in cancer, DNA damage, and cell cycle progression.  The information presented here does not fit the theme of HIV-1 macrophage infection and reverse transcriptase.  In addition, the concepts discussed in lines 656-673 were already covered in the review article mentioned in my previous comment.

  • We thank the reviewer for their comments. We included this section as many updates have occurred in recent years on the impact of SAMHD1 outside of retroviral restriction and we felt leaving this information out would leave an incomplete picture of the SAMHD1 field. We agree this shifts the focus of the review and have made major cuts to this section in order to maintain focus.

Reviewer 2 Report

This review describing the Interplays between HIV-1 Reverse Transcriptase 2 Enzyme Kinetics and Host SAMHD1 Protein is overall extensive and is well written. I have a general comment:

1. The authors have not mentioned  the role of SAMHD1 in resting T cells. I believe its relevant in section 8.2

Author Response

Reviewer 2

1) The authors have not mentioned the role of SAMHD1 in resting T cells. I believe its relevant in section 8.2. 

  • We thank the reviewer for this feedback. We agree this is a relevant addition and have added information about the role of SAMHD1 in resting T-cells and Dendritic Cells into section 8.2 (now section 8).

Reviewer 3 Report

The review article by Oo, Bowen, and Kim on SAMHD1 and HIV-1 RT comprehensively capture various roles and impacts SAMHD1 which is timely and valuable. In fact, the review broadly covers SAMHD1 beyond its impact on RT which is primarily by reducing dNTP concentration.

Minor comments:

1.     Title and abstract do not reflect the broader scope of the review article beyond HIV.

2.     Adding a figure to visualize the structural details discussed in section 7 can be useful.

3.  Line 107 reads that M184 is highly conserved. Later in the manuscript, the authors discuss M184V as a resistance mutation. The authors need to clarify this.

Author Response

Reviewer 3

1) Title and abstract do not reflect the broader scope of the review article beyond HIV.

  • We thank the reviewer for their remarks. We have cut down on the section that discusses the broader scope of SAMHD1, RT inhibitors, and antiviral resistance in order to help maintain to focus of the article.

2) Adding a figure to visualize the structural details discussed in section 7 can be useful.

  • We thank the reviewer for their comments. We agree it would be useful to visualize the structure details discussed in section 7 and have added in a figure demonstrating SAMHD1 quaternary structure and where regulatory amino acids are located (Figure 2).

3) Line 107 reads that M184 is highly conserved. Later in the manuscript, the authors discuss M184V as a resistance mutation. The authors need to clarify this.

  • We thank the reviewer for this suggestion. We have removed the sections discussing RT inhibitors and drug resistance in order to preserve the focus of the review. Therefore, we no longer discuss the M184V resistance mutation in this revision.

Round 2

Reviewer 1 Report

1) Line 581 - the word "highly" should be removed

2) References section: journal abbreviations - The references need to be reformatted from #22 onward.

Author Response

Please see the uploaded reponsex2. Thank you.
